# Halotolerance of Phytoplankton and Invasion Success of Nostocalean Cyanobacteria Under Freshwater Salinization

**DOI:** 10.3390/microorganisms13061378

**Published:** 2025-06-13

**Authors:** Izabelė Šuikaitė, Gabrielė Šiurkutė, Robert Ptacnik, Judita Koreivienė

**Affiliations:** 1Laboratory of Algology and Microorganisms Ecology, State Scientific Research Institute Nature Research Centre, Akademijos St. 2, LT-08412 Vilnius, Lithuania; gabriele.siurkute@gamtc.lt; 2Faculty of Technology, Natural Sciences and Maritime Sciences, University of South-Eastern Norway, Lærerskoleveien 40, 3679 Notodden, Norway; robert.ptacnik@usn.no; 3WasserCluster Lunz, Dr. Kupelwieser-Prom. 5, 3293 Lunz am See, Austria

**Keywords:** microbial invasion, cyanobacteria, halotolerance, alien species, harmful algae, disturbed ecosystems, freshwater salinization

## Abstract

Disturbed ecosystems are particularly susceptible to biological invasions. Increasing freshwater salinization, caused by anthropogenic factors, can alter the phytoplankton community and favour newly arrived halotolerant species. This study investigates the halotolerance of four Nostocalean cyanobacterial species—the native to Europe, *Aphanizomenon gracile*, and alien *Chrysosporum bergii*, *Cuspidothrix issatschenkoi*, and *Sphaerospermopsis aphanizomenoides*—using monoculture experiments under varying NaCl concentrations. Additionally, we performed two microcosm experiments to explore shifts in biodiversity in freshwater phytoplankton communities sourced from artificial reservoirs and assess their susceptibility to cyanobacterial invasion under salinity stress. Results showed that all Nostocalean cyanobacteria were halotolerant under mild salinities (up to 1 g/L NaCl), with *Chrysosporum bergii* and *Sphaerospermopsis aphanizomenoides* demonstrating the most salt tolerance. In the microcosm experiment, changes in community composition were driven by the halotolerance of dominant groups. Water body 1, dominated by Bacillariophytina, reduced its biomass of phytoplankton at high salinity (5 g/L NaCl), while water body 2, dominated by Chlorophytina, remained stable regardless of disturbance. Both cyanobacteria successfully invaded both halotolerant and halosensitive communities, increasing their dominance as salinity rose. Our findings suggest that anthropogenic stressors such as freshwater salinization can alter the phytoplankton community and increase a competitive advantage to certain taxa, including widespread alien cyanobacteria, potentially promoting invasions and bloom formation.

## 1. Introduction

Biological invasions are widely recognized as one of the leading threats to biodiversity in the Anthropocene and are expected to increase in the future [1]. Global trade and human mobility are major drivers of invasions. The success of species entering non-native habitats depends on ecological interactions between native and alien species, as well as environmental characteristics that can alter biodiversity and increase the risk of species extinction [2]. Over half a century ago, scientists recognized that disturbed ecosystems are particularly susceptible to invasions [3,4]. Alien species tend to be introduced more frequently in urban areas, where they can take advantage of resources found in these disturbed environments [5,6]. Freshwater ecosystems, especially artificial ponds, are particularly vulnerable to biodiversity changes and invasions due to both climate change and anthropogenic pressures [7,8,9]. These ecosystems are subjected to multiple stressors from agricultural and urban land use, including nutrient runoff that promotes eutrophication and the introduction of harmful pollutants such as pesticides and salts [10,11].

As a result of droughts [12], agriculture [13], mining, the use of road deicers [14], and the incursion of coastal waters, the salinization of freshwater ecosystems has become a global challenge for aquatic habitats [15]. Increased freshwater salinization threatens the biodiversity and functioning of freshwater ecosystems [16] and causes a drastic shift in plankton communities, even with small changes in salinity levels, which are often below the general chloride (Cl^−^) threshold guidelines [17,18,19]. Increased salinity selectively pressures less salt-tolerant native species, allowing more resilient and halotolerant species to dominate.

Nostocalean cyanobacteria have gained considerable interest due to their rapid spread and establishment in new environments, often leading to harmful algal blooms. Increased water temperatures and eutrophication are among the factors benefiting their establishment [20]. However, there are only a few studies addressing the potential effects of salinization on the development of freshwater Nostocalean cyanobacteria, most of which focus on non-European strains. Nevertheless, these studies indicate that cyanobacteria, including some alien species, can tolerate saline environments [21,22,23,24]. Under freshwater salinization, cyanobacteria may gain a competitive advantage over eukaryotic organisms, allowing them to dominate [25]. Additionally, the production of cyanotoxins may be influenced by salinization and is thought to be an adaptive mechanism to counteract salt stress [26,27]. Consequently, the risk of toxic cyanobacterial blooms may increase due to the ongoing salinization of freshwater ecosystems.

The aim of this study was to examine how increasing salinity influences the growth of native and alien Nostocalean cyanobacteria and to assess the impact of salinization on freshwater phytoplankton community structure and its susceptibility to cyanobacterial invasion. Therefore, we conducted a monoculture experiment using four species of cyanobacteria, a native *Aphanizomenon gracile* to European freshwaters and alien *Chrysosporum bergii*, *Cuspidothrix issatschenkoi*, and *Sphaerospermopsis aphanizomenoides*, to assess their halotolerance under varying levels of salinization. Additionally, we performed two microcosm experiments to evaluate the biodiversity shifts in natural freshwater phytoplankton communities from two mesotrophic, oligohaline artificial reservoirs exposed to salinization stress and examine their susceptibility to cyanobacterial invasion. We hypothesized that the development of phytoplankton communities is primarily conditioned by changes in salinity and that halotolerant cyanobacteria would show success under the disturbance of salinization. 

## 2. Materials and Methods

### 2.1. Cyanobacterial Isolation and Maintenance

The isolates of *Aphanizomenon gracile*, *Chrysosporum bergii*, *Cuspidothrix issatschenkoi*, and *Sphaerospermopsis aphanizomenoides* used in this study are listed in Table 1. 13 isolates were sourced from freshwater lakes Jieznas (54°59′27.01″, 24°18′03.96″), Gineitiškės (54°73′79.55″, 25°18′53.21″), Rėkyva (55°86′52.86″, 23°30′13.16″), and Simnas (54°39′95.36″, 23°63′83.37″) in Lithuania and from Hostivař Reservoir (50°03′90.06″, 14°54′00.39″) in the Czech Republic during midsummer. The single filament isolates were retrieved from the lake water samples using a glass microcapillary pipette, grown in MWC medium, and stored in the culture collection at the Nature Research Centre (Vilnius, Lithuania) at illumination of 100 μmol photons m^−2^ s^−1^ under a 16:8 light photoperiod regime at 20 °C. The experiments were carried out at WasserCluster Lunz (Lunz am See, Austria), where the isolates were acclimated to local laboratory conditions—grown in a medium consisting of 10% WC and 90% sterile lake water, maintained under 100 μmol photons m^−2^ s^−1^ illumination, with a 16:8 light–dark photoperiod at 18 °C—and kept under these conditions throughout the experiment. All isolates were identified based on their morphological characteristics, following the descriptions provided by Komárek 2013 [28].

### 2.2. Monocultures Experiment

The strains of *Aphanizomenon gracile*, *Chrysosporum bergii*, *Cuspidothrix issatschenkoii*, and *Sphaerospermopsis aphanizomenoides* were used for halotolerance monoculture experiments across a range of salinity levels using sodium chloride (NaCl) in 40 mL culture flasks. We used 0, 1, 2.5, 5, and 10 g/L NaCl concentrations for *C. bergii* and *S. aphanizomenoides*, whereas *A. gracile* and *C. issatschenkoi* were tested only up to 5 g/L, as they did not grow at 10 g/L. Each experimental condition was set up in triplicate. Initial cultures in the exponential growth phase were used as inoculum for new batch cultures, with salinity levels gradually adjusted over a 5-day incubation period. After this adjustment, measurements were taken for the following 3 days using a fluorometer Varioskan Flash Multiplate reader (Thermo Fisher Scientific, Waltham, MA, USA), making the total experiment duration 8 days. Fluorometric readings were taken using 96-well microtiter plates and were dark-incubated for 30 min prior to measurements.

Each culture in the exponential growth phase was monitored by phycocyanin fluorescence (excitation wavelength at 630 nm, emission at 660 nm), with results recorded in relative fluorescence units (RFU). Growth rates (μ) were calculated for the exponential phase using the equation:μ = ln (N_t_/N_0_)/∆t
where N_0_ and N_t_-RFU are values at the beginning and the end of the exponential growth phase, and ∆t is the period of the exponential phase expressed in days [29].

### 2.3. Microcosm Experiment

The microcosm experiments were conducted in a walk-in environmental chamber at an illumination of 100 μmol photons m^−2^ s^−1^ under a 12:12 light photoperiod regime at 18 °C. 0.5 L tissue culture flasks were filled with surface water collected from two gravel pit lakes near Petzenkirchen, Lower Austria (water body 1: 48°15′47.10″, 15°17′39.14″; water body 2: 48°15′4379″, 15°17′62.98″) collected in May 2023. Water was filtered through a 100 μm mesh sieve to remove large zooplankton. The total phosphorus concentration was 22.75 μg/L for water body 1 and 20.18 μg/L for water body 2, classifying the water bodies as mesotrophic [30]. The experiment included two sets of culture flasks: one with the native phytoplankton community (control group) and another in combination with the native phytoplankton community with the invasive species inoculated at the beginning of the experiment (invaded group). Based on the salt tolerance experiment, the most halotolerant species were selected for the invasion experiment: *Chrysosporum bergii* (isolate NRC/GIN/2017/D6) for water body 1 and *Sphaerospermopsis aphanizomenoides* (isolate NRC/SIM/2022/A1) for water body 2. Four salinity levels (0, 0.2, 1, and 5 g/L NaCl) were adjusted over 3 days. These salinity variations would correspond to a transfer from freshwater (0 and 0.2 g/L NaCl) to the oligohaline level (1 and 5 g/L NaCl) [31]. Each condition was replicated in triplicate, resulting in 24 flasks per water body. After 3 days of acclimatization, the experiment continued for an additional 10 days, making a total of 13 experimental days. The initial biovolume of the invaders was measured using autofluorescence and later verified through microscopy, which accounted for 25–26% of the total phytoplankton biovolume.

Samples for phytoplankton analysis (0.5 L) were fixed with neutral Lugol’s solution. The preserved samples were decanted to approximately a volume of 10 mL prior to counting after they had been allowed to settle for at least 7 days. Phytoplankton species identification and counting were performed using a Nageotte counting chamber under a light microscope [32]. At least 400 cells per sample were estimated. Biovolume was calculated based on the cell numbers and mean cell volumes of species using formulas for geometric shapes [33]. Biomass was estimated from cell counts in a known chamber volume [34] and expressed as a percentage of total phytoplankton biomass. To express this biomass as a percentage, the calculated biomass is multiplied by 100 and divided by the total biomass of all phytoplankton groups in one sample. Taxonomic identification of phytoplankton species was based on morphology according to the descriptions of several books [28,35,36,37,38,39,40,41,42], and their currently accepted taxonomic names were confirmed according to Algaebase [43].

### 2.4. Data Visualization and Statistical Analysis

Statistical analysis and data plotting were performed using software R (version 3.5.2), with a significance level set at *p* < 0.05. A Mann–Whitney U test was used to compare two data sets. For comparisons of more than two groups, either an ANOVA for normally distributed data or a Kruskal–Wallis test (as a non-parametric alternative) was applied using the *dplyr* and *car* packages. Pairwise multiple comparisons were made using Dunn’s test for non-parametric data and Tukey’s post-hoc test for ANOVA in the *dunn.test* and *multcomp* packages, respectively. The normality of the data was tested using the Shapiro–Wilk test, and the homogeneity of variance was assessed using Levene’s test. To evaluate differences in community structure (in biomass) between control and invaded groups, PERMANOVA was performed using the *vegan* package. Both PERMANOVA and one-way ANOVA were used to compare the total biomass of phytoplankton among salinity levels in control and invaded groups, excluding the invader, in order to compare the response of the native community. Data visualization of the growth rates of cyanobacterial isolates and the community structure from the microcosm experiment was carried out using the *ggplot2*, *ggpubr*, *ggpattern*, *patchwork*, and *dplyr* packages. Values of growth rates and biomass are represented as means with standard deviations among replicates.

## 3. Results

### 3.1. Monocultures Experiment

The growth rates of tested cyanobacterial isolates under different salinity levels are illustrated in Figure 1. The maximum mean growth rates (μ_max_) varied among the species. Most isolates for *Sphaerospermopsis aphanizomenoides* had the μ_max_ of around 1 d^−1^ at salinities between 0 and 2.5 g/L NaCl, except for one isolate, which reached the μ_max_ of 1.318 ± 0.06 d^−1^ at 0 g/L NaCl. Two isolates of *Cuspidothrix issatschenkoi* reached their μ_max_ at 0 salinity level (μ_max_ = 1.076 ± 0.16 d^−1^). Lower μ_max_ values were observed for two isolates of *Aphanizomenon gracile*; for one isolate, μ_max_ was detected at 0 salinity level (μ_max_ = 0.873 ± 0.19 d^−1^), and for another isolate, it was similar both at 0 and 1 g/L NaCl (μ_max_ = 0.743 ± 0.04 d^−1^ and μ_max_ = 0.757 ± 0.05 d^−1^, respectively). The lowest values of μ_max_ were reported for isolates of *Chrysosporum bergii*, which varied considerably among isolates, reaching μ_max_ at salinities of 1 and 2.5 g/L NaCl (μ_max_ = 0.614 ± 0.03 d^−1^).

The growth rates of *Sphaerospermopsis aphanizomenoides* under varied salinity levels differed significantly from those of the other species (one-way ANOVA F = 18.47, df = 3, *p* < 0.001; post-hoc Tukey’s test *p* < 0.05). Statistical analysis indicates that salinity levels had a significant effect on the growth rates of all tested isolates (*p* < 0.001) (Table 2). Isolates of *Aphanizomenoides gracile* showed positive growth rates up to 2.5 g/L of NaCl and were suppressed at 5 g/L of NaCl. One isolate of *Cuspidothrix issatschenkoi* showed negative growth rates already at 2.5 g/L of NaCl, while another isolate showed slightly positive growth at 2.5 g/L of NaCl (μ = 0.087 ± 0.17 d^−1^) and was suppressed at 5 g/L NaCl. In contrast, isolates of *Chrysosporum bergii* and *S. aphanizomenoides* demonstrated wider halotolerance, sustaining positive growth rates up to 10 g/L of NaCl with the highest value of 0.220 ± 0.03 d^−1^ and 0.523 ± 0.18 d^−1^, respectively. No significant differences were observed among isolates of *A. gracile*, *C. issatschenkoi*, and *S. aphanizomenoides*, except for *C. bergii* isolate (p = 0.004). Dunn’s post-hoc test revealed significant differences in salt tolerance among the two *C. bergii* isolates NRC/GIN/2017/F6 and NRC/KYV/2015/E2 (*p* < 0.001). Significant differences were observed at 0 g/L NaCl among isolates (Kruskal–Wallis: χ² = 10.23, *p* = 0.037), with two isolates from Lakes Jieznas and Rėkyva (NRC/JIE/2015/E2 and NRC/KYV/2015/E2) showing lower growth rates (μ = 0.25 ± 0.02 d^−1^) compared to isolates from Lake Gineitiškės (NRC/GIN/2017/B3, NRC/GIN/2017/D6, and NRC/GIN/2017/F6), which had higher rates (μ = 0.51 ± 0.10 d^−1^), indicating growth rate variations among isolates from different lakes.

### 3.2. Microcosm Experiment

Phytoplankton assemblages of microcosm experiment of two water bodies are represented by taxonomic groups in Figure 2. On the first day of the experiment, the initial control phytoplankton community in water body 1 was predominantly composed of Myzozoa (the highest biomass of *Ceratium*) that accounted for 42% of total phytoplankton biomass, with subdominant groups including Bacillariophytina (*Cyclotella*) that accounted for 21%, Chlorophytina (*Tetraedron*, *Scenedesmus*, *Nephrochlamys*, *Monoraphidium*, *Oocystis*, unidentified species from Chlorellales order) that accounted for 20%, and Ochrophytina (unidentified) that accounted for 15%. The total biomass of the control phytoplankton community was measured at 2.37 mg/L. After inoculation with 0.83 mg/L of *Chrysosporum bergii* culture, the invader contributed 26% to the total phytoplankton biomass.

On the first day of the experiment in water body 2, the dominant phylum in the initial control phytoplankton community was Chlorophytina (*Tetraedron*, *Pediastrum*, *Monoraphidium*, *Scenedesmus*, unidentified species from Chlorellales order) with 35% of the total phytoplankton biomass. Subsequently, the subdominant groups were Ochrophytina (unidentified), Cyanobacteria (*Aphanocapsa*, *Chroococcus*, *Cyanodictyon*, *Planktolyngbya*), and Bacillariophytina (*Cyclotella*, *Nitzschia*) corresponding to 27%, 21% and 15%, respectively. The total biomass of the control phytoplankton community was 1.63 mg/L. After adding 0.54 mg/L of *Sphaerospermopsis aphanizomenoides* culture, the invader contributed 25% to the total phytoplankton biomass.

By the end of the experiment, the biomass of native phytoplankton communities tended to decrease compared to the biomass of their initial communities of both water bodies (Figure 2). There was no significant difference in native communities’ structure between control and invaded groups in water body 1 (PERMANOVA: F = 0.76; R^2^ = 0.0350; *p* = 0.452) and water body 2 (PERMANOVA: F = 1.50; R^2^ = 0.0697; *p* = 0.193).

In water body 1, the biomass of native communities significantly differed in both control and invaded groups among salinity levels at the end of the experiment (one-way ANOVA: F = 10.64, *p* = 0.001; F = 18.16, *p* = 0.001, respectively). The highest biomass of the native community was observed at 0.2 g/L NaCl, reaching 2.55 ± 0.5 mg/L for the control group and 2.40 ± 0.5 mg/L for the invaded group. At the end of the experiment, the communities were primarily composed of Bacillariophytina at all salinity levels, with the maximum value of 1.35 ± 0.2 mg/L in the control group and of 1.52 ± 0.5 mg/L in the invaded group at 0.2 g/L NaCl and were sharply reduced at 5 g/L NaCl to 0.28 ± 0.2 mg/L and 0.34 ± 0.1 mg/L, respectively. Ochrophytina was in the subdominant group at 0 and 0.2 g/L NaCl and reached maximum values of 0.81 ± 0.3 mg/L and 0.58 ± 0.1 mg/L at 0.2 g/L NaCl salinity level in the control and invaded groups, respectively, but was absent at 1 and 5 g/L NaCl. Statistically significant differences among salinity levels were seen for Bacillariophytina, Coscinodiscophytina and Ochrophytina (Table 3).

In contrast, at the end of the experiment in water body 2, the biomass of native phytoplankton communities did not differ among salinity levels in both control and invaded groups (one-way ANOVA: F = 4.28, *p* = 0.74; Kruskal–Wallis: χ^2^ = 2.12 *p* = 0.548, respectively). The native phytoplankton communities were primarily dominated by Chlorophytina, whose biomass remained similar across all salinity levels (varied from 0.65 ± 0.1 mg/L to 1.11 ± 0.7 mg/L) (Figure 2). The biomass of Chlorophytina, together with Charophyta and Myzozoa, did not show significant variation across salinity levels (Table 3). Bacillariophytina and Ochrophytina made up a smaller proportion of the total phytoplankton biomass than in water body 1 and were reduced at 5 g/L NaCl (Table 3) (post-hoc Tukey’s test: *p* < 0.05; post-hoc Dunn’s test: *p* < 0.05, respectively). In contrast to water body 1, the biomass of subdominant group Cyanobacteria varied significantly between salinity levels, reaching the highest values at 5 g/L NaCl up to 0.17 ± 0.4 mg/L and 0.18 ± 0.0 mg/L in control and invaded communities, respectively (Table 3) (post-hoc Tukey’s test: *p* < 0.05).

In water body 1, the initial biomass of *Chrysosporum bergii* (0.83 mg/L) changed only slightly across all salinity levels (Figure 2). A positive change in biomass was observed only at 1 g/L NaCl by the end of the experiment, where biomass slightly increased to 0.87 ± 0.1 mg/L. *C. bergii* biomass showed only a marginally significant difference among salinity treatments (one-way ANOVA: F = 4.35, *p* = 0.05), suggesting a negligible effect of salinity on its biomass change. The invader’s contribution to total phytoplankton biomass declined slightly at 0 and 0.2 g/L NaCl (reaching 19.44 ± 5.0% and 22.09 ± 7.0%, respectively) compared to its contribution on the first day (26%) (Figure 3). In contrast, at 1 and 5 g/L NaCl, due to reduced biomass of the native community, *C. bergii* increased its dominance to 33.67 ± 19.8% and 51.33 ± 2.2%, respectively. Notably, the contribution at 5 g/L NaCl was significantly higher than at all other salinity levels (one-way ANOVA: F = 28.3, *p* < 0.001; post-hoc Tukey’s test: *p* < 0.001).

In water body 2, the initial biomass of *Sphaerospermopsis aphanizomenoides* (0.54 mg/L) increased considerably with rising salinity, reaching a maximum at 5 g/L NaCl, with a fivefold increase in biomass (2.68 ± 0.3 mg/L) (one-way ANOVA: F = 29.82, *p* < 0.001; post-hoc Tukey’s test: *p* < 0.05). The relative contribution of the invader also showed a gradual increase with salinity, peaking at 70.19 ± 3.8% at 5 g/L NaCl. However, this increase was not statistically significant among salinity levels (one-way ANOVA: F = 3.91, *p* = 0.062).

## 4. Discussion

Freshwater salinization is increasingly recognized as a major environmental threat, driven by both climatic changes and anthropogenic pressures. Small and artificial water bodies are particularly vulnerable as they are more susceptible to water loss through evaporation and are often located in urban or agricultural landscapes, where ion accumulation from runoff can be intensified. This study explores the halotolerance of the widely spreading Nostocales cyanobacteria, both alien and native to Europe. Additionally, we demonstrated that anthropogenic salinization can alter phytoplankton community structure, depending on the salt tolerance of dominating taxonomic groups, increase the potential for successful invasions, and promote cyanobacterial blooms. This study is the first, to our knowledge, to investigate the halotolerance and invasion potential of alien Nostocalean cyanobacteria in Europe under salinity stress—thus offering new insights into invasion dynamics under ongoing global change.

### 4.1. Halotolerance of Nostocalean Cyanobacteria

The ability of Nostocalean cyanobacteria to adapt under elevated salinity conditions remains poorly understood. Understanding the adaptive potential of alien cyanobacteria under such conditions is crucial for predicting their establishment and ecological impact in new environments under ongoing salinization. Therefore, we performed a monoculture experiment using freshwater *Nostocalean* cyanobacteria, a native *Aphanizomenon gracile* to European freshwaters and alien *Chrysosporum bergii*, *Cuspidothrix issatschenkoi*, and *Sphaerospermopsis aphanizomenoides*, to assess their halotolerance at both species and strain levels. The response of 13 isolates to salinity stress varied among species (Figure 1), suggesting different abilities to cope with salinity changes. Among the species tested, two alien species, *C. bergii* and *S. aphanizomenoides*, showed the broadest halotolerance, maintaining positive growth rates up to 10 g/L NaCl for some isolates referring to a mesohaline environment [30].

The halotolerance of *C. bergii* is not surprising as it is a species native to brackish water ecosystems in the Ponto-Caspian region and, therefore, considered halophilic [44]. It has never been reported to form blooms in freshwater nor in brackish waters [45], which may be attributed to its low growth rates across all salinity levels shown in this study (Figure 1). There is a lack of studies examining how this species responds to different salinity levels, with the exception of a study for one non-European isolate (Senegal) from Duval et al. [24], who, similar to some isolates from our study, reported growth suppression at 10 g/L NaCl, although showed an optimum at 0 g/L NaCl (μ_max_ = 0.580 ± 0.02 d^−1^). In our study, *C. bergii* showed a varied response among isolates (Table 2), with growth patterns appearing to reflect their habitat of origin. For instance, isolates from Lakes Jieznas and Rėkyva showed lower growth rates at 0 g/L NaCl compared to those from Lake Gineitiškės, although all isolates peaked at 1–2.5 g/L NaCl, reaching a maximum of 0.614 ± 0.03 d^−1^. These results suggest that *C. bergii* is a halotolerant species; however, some strains may not prefer strictly freshwater conditions and instead perform better with moderate salinity increases.

Although *Sphaerospermopsis aphanizomenoides* is considered an originally freshwater species [46], it showed a unique growth response under increased NaCl concentrations (Figure 1). *S. aphanizomenoides* is commonly found in high densities in both freshwater and brackish environments [45,47], which aligns with its highest growth rates observed in our study across all salinity levels (μₘₐₓ = 1.318 ± 0.06 d^−1^ at 0 g/L NaCl). Moisander et al. [21] reported comparable findings to our study, showing high growth rates of up to 10 g/L NaCl for a strain isolated from the USA. Moreover, there were no significant differences among the four isolates from Lithuania and the Czech Republic, suggesting a similar intraspecific response (Figure 1). These results indicate that *S. aphanizomenoides* has a strong capacity for development across a broad salinity range and the ability to form blooms in both freshwater and brackish waters.

Alien *Cuspidothrix issatschenkoi* and native *Aphanizomenon gracile* were shown as comparatively halosensitive species in our study. Although *A. gracile* and *C. issatchenkoi* have been found in brackish waters on a few occasions [44,48,49,50], bloom formation by these species has only been reported in freshwater environments [51,52,53]. This is consistent with our findings, where two isolates of *C. issatschenkoi* were the most sensitive to salinization stress, showing high growth rates at 0 g/L NaCl (μ_max_ = 1.076 ± 0.16 d^−1^) but ceased growth at 1 g/L NaCl. Similarly, *Aphanizomenon gracile*, a species native to Europe, showed moderate halotolerance, maintaining positive growth rates up to 2.5 g/L NaCl but were suppressed at 5 g/L NaCl (Figure 1). Our findings are in partial agreement with a previous study showing that a European strain of *A. gracile* failed to increase biomass after the addition of 2 g/L NaCl, suggesting it may not survive beyond oligohaline conditions (> 5 g/L NaCl) [54]. This suggests that while both species can be found in brackish environments, their potential to thrive under such conditions is limited.

Exposure to NaCl resulted in reduced growth rates across all species, although all were able to tolerate salinity levels up to 1 g/L of NaCl (equivalent to 0.6 g/L Cl^−^), which exceeds the general chloride threshold guidelines (0.12–0.25 g/L Cl^−^) [17] or that can be found in freshwater lakes affected by road salt contamination [14]. Furthermore, our findings suggest that certain genera or isolates of cyanobacteria show greater tolerance to high salinity fluctuations, comparable to mesohaline conditions (>5 g/L NaCl) [30], providing them with a competitive advantage in estuaries or during extreme salinization events, such as those caused by urban runoff, where eukaryotic phytoplankton may not succeed [55]. However, to better understand the extent of intraspecific variability of the studied species, these findings should be confirmed with a larger number of strains and under diverse environmental conditions.

### 4.2. Effects of Salinization on Phytoplankton Community Composition

We conducted microcosm experiments using freshwater phytoplankton communities from two artificial water reservoirs to investigate the effects of salinization on phytoplankton communities. Our results revealed distinct responses to salinization in native communities from the two water bodies. At the end of the experiment in water body 1, where Bacillariophytina dominated under all conditions, the native phytoplankton community showed sensitivity to salinization, with the dominant group significantly reduced at higher salinities (1 and 5 g/L NaCl) (Figure 2) and (Table 3). In contrast, the phytoplankton community of water body 2 remained halotolerant, with stable biomass across salinity treatments, primarily due to the dominance of Chlorophytina, which showed resilience under all salinity levels in both water bodies (Figure 2) and (Table 3). The results suggest that the stability of phytoplankton communities under salinization stress depends on the halotolerance of the dominant species. Additionally, we did not observe compensatory growth from rare halotolerant taxa after the dominant species were lost due to the disturbance, which led to a reduction in phytoplankton biomass in water body 1 [23,56].

Moreover, at 0.2 g/L NaCl in water body 1, the peak in total phytoplankton biomass was mainly due to the increased abundance of Ochrophytina and Bacillariophytina groups (Figure 2), likely driven by their preference for mild salinization. However, these groups were either lost or experienced lower growth under increased salinity stress (1 and 5 g/L NaCl). Similarly to our study, Floder et al. [23] reported a suppression of the member from the Ochrophytina group at 0.5 g/L of artificial sea salt. In contrast, a study by Astrog et al. [56] found an increase in Ochrophytina abundance with rising salt concentrations (>0.64 g/L, equivalent > 1.1 g/L NaCl), supporting the idea of species-specific salinity responses. However, we did not identify Ochrophytina to the genus or species level. In our study, members of the Bacillariophytina group, primarily consisting of *Cyclotella*, *Nitzschia*, and *Synedra*, showed growth suppression at 1 and 5 g/L NaCl. In contrast, Floder et al. [23] demonstrated that *Nitzschia* and *Synedra* increased their development within a salinity range of 0–2 g/L NaCl but declined above a salinity of 2 g/L. *Cyclotella*, however, showed no salinity preferences up to 5 g/L of salt and developed a stable biomass [23]. Similarly, the contrasting responses of the same genera to increasing salinity may be due to the species-specific behaviours or the presence of different ecotypes [57].

We hypothesized that salinity changes would influence phytoplankton community development. Indeed, our results show that while salinity clearly altered community composition in water body 1 (Figure 2), no significant change in total phytoplankton biomass was observed in water body 2—even at mesohaline levels (5 g/L NaCl). This resilience was primarily due to the dominance of Chlorophytina taxa, which maintained stable growth across all salinity treatments. Although the salinity tolerance of freshwater Chlorophytina—particularly the genus Scenedesmus, which was prevalent in our study (Appendix A)—has been demonstrated largely in laboratory settings [58,59,60], our findings confirm this tolerance in natural community assemblages. While previous studies have shown that even small salinity increases can shift freshwater phytoplankton structure [61,62,63], our results extend this by demonstrating that some taxa can persist even under elevated salinity, suggesting their role in stabilizing community structure during salinization events.

### 4.3. Invasion Success of Halotolerant Cyanobacteria

The benefits of increased freshwater salinization for cyanobacteria have been demonstrated in several studies [64,65,66]. These studies suggest that cyanobacteria can overtake eukaryotic phytoplankton species, leading to cyanobacterial blooms under increased salinity conditions [67]. In the microcosm experiment of this study, both water bodies were initially dominated by native communities with low cyanobacterial biomass, which did not result in cyanobacterial blooms (Figure 2). However, the absence of native cyanobacterial competitors likely facilitated the successful establishment of the two invading cyanobacterial species, as there were no members occupying the same ecological niche, reducing competition for resources [68]. We hypothesize that newly arrived halotolerant cyanobacteria would show success under the disturbance of salinization. Our monoculture experiment demonstrated significant halotolerance in cyanobacteria, which led us to select two isolates of the best-performing species, *Chrysosporum bergii* and *Sphaerospermopsis aphanizomenoides*. Their broad halotolerance was further confirmed through the invasion experiment.

Although both invasive cyanobacteria increased their relative biomass with rising NaCl concentrations, their biomass production differed, highlighting two distinct pathways through which newly introduced halotolerant cyanobacteria can enhance their contribution. In water body 1, *Chrysosporum bergii* maintained stable biomass across all salinity levels, giving it a competitive advantage under higher salinities due to the lack of resilience in the native community (Figure 2). As observed in the monoculture experiments, these results suggest that *C. bergii* has broad halotolerance, but its slow growth rates limit its biomass development (Figure 1). There is limited research on how *C. bergii* behaves in brackish waters, but this study suggests that its broad halotolerance could facilitate its spread into more saline environments, providing a competitive advantage over salt-sensitive species and enabling its successful establishment. The ability to maintain stable biomass and the absence of competitors at higher salinities could contribute to its long-term establishment.

In contrast, in water body 2, although the native community remained stable across all salinity levels, *Sphaerospermopsis aphanizomenoides* showed a clear preference for higher salinities. As salinity increased, its biomass grew progressively, allowing it to outcompete the native species and establish itself. These results align with those from the monoculture experiment, demonstrating *S. aphanizomenoides’* broad halotolerance and rapid growth rates (Figure 1). Notably, Sabour et al. [47] reported the occurrence of *S. aphanizomenoides* in a brackish lake with an average salinity of 6.6 g/L, where it formed a dense bloom, comprising 94% of the total phytoplankton biomass in a hypertrophic lake in Morocco. These findings highlight the ability of *S. aphanizomenoides* to form strong blooms under mesohaline conditions.

Notably, the presence of the invaders did not impact the native phytoplankton community in our study, as no significant differences were observed in the community structure between the control and invaded native phytoplankton groups under all salinities. These findings contrast with those of Weithoff et al. [69], where the temporal appearance of the cyanobacterium *Raphidiopsis raciborskii* led to significant changes in community structure. Similarly, Buchberger and Stockenreiter [70] demonstrated unsuccessful invasions by three taxonomic groups (Chlorophyta, Cyanobacteria, and Bacillariophyta), which nonetheless affected the resident phytoplankton community by increasing its diversity. Notably, the effects of invaders on phytoplankton communities may be obscured at higher taxonomic levels, as shown by Buchberger [70]. Species-level identification in our study could reveal more detailed effects.

This study demonstrates a successful invasion of freshwater Nostocalean cyanobacteria under salinization stress. Although there are no prior reports of cyanobacterial invasion under salinity stress, previous studies of cyanobacterial invasion without disturbance factors often showed unsuccessful outcomes [69,70]. The failure observed in these studies has been attributed to the inability of cyanobacteria to withstand the transfer from culture to lake water. However, in our study, both invader species maintained high biomass levels throughout the experiment, demonstrating significant success. Additionally, the successful invasion in our experiment can be explained by the high propagule pressure (25–26% of total phytoplankton biomass), compared to Buchberger’s study, where it was only 10%. Propagule pressure is recognized as a key factor influencing the success of invasions [71]. Moreover, our study demonstrated that the disturbance factor of salinization can facilitate invasion success. The findings support key ecological invasion theories [3,4], which posit that ecological disruptions weaken competitive resistance and create opportunities for invasive taxa.

Environmental stressors extend beyond salinization to include warming and eutrophication, all of which have been shown to decrease biodiversity, alter competitive hierarchies, and amplify invasion success [20]. Microbial invasions could have cascading consequences for biodiversity, nutrient cycling, and food web stability. One of the greatest risks to aquatic life is cyanobacterial blooms, which reduce light availability, inhibit the growth of other primary producers, contribute to hypoxia through decomposition, and produce toxins that compromise recreational water use and drinking water supplies [72]. The study highlights the importance of monitoring not only nutrient loads but also ion concentrations—especially those elevated by urbanisation, which increases runoff and the use of deicing salts. Climate change further complicates these dynamics, as altered precipitation patterns, increased evapotranspiration, and rising temperatures may intensify blooms. As these pressures rise, more ecosystems may cross critical thresholds, making microbial invasions not only possible but increasingly probable.

## 5. Conclusions

The study reveals that freshwater phytoplankton communities show variation in resilience to increasing salinity, primarily influenced by the behaviour of the dominant taxa. While salinity could alter phytoplankton community composition, a community dominated by Chlorophytina maintained stability, demonstrating tolerance even at mesohaline levels. Similarly, Nostocalean cyanobacteria tolerated mild salinization, with two alien species—*Chrysosporum bergii* and *Sphaerospermopsis aphanizomenoides*—achieving high growth rates under mesohaline conditions. In the microcosm experiment, both halotolerant invading cyanobacteria benefited from rising salinity, increasing their dominance. These findings suggest that anthropogenic salinization of inland urban waters can act as a disturbance, disrupting native assemblages and enhancing the success of halotolerant taxa, including non-native bloom-forming Nostocalean cyanobacteria—thereby posing a significant threat to ecosystem health. Ultimately, this study highlights the need to consider salinization as a key factor in the management and conservation of freshwater ecosystems, especially in the face of increasing anthropogenic pressures and climate-induced changes.

## Figures and Tables

**Figure 1 microorganisms-13-01378-f001:**
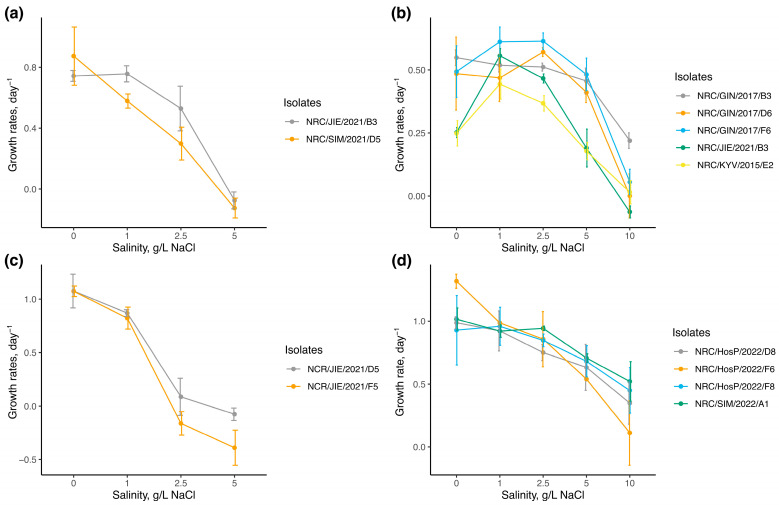
Growth rates of cyanobacterial isolates across a range of different salinity levels (n = 3, for each salinity condition) in cultures of (**a**)—*Aphanizomenon gracile*, (**b**)—*Chrysosporum bergii*, (**c**)—*Cuspidothrix issatschenkoi*, (**d**)—*Sphaerospermopsis aphanizomenoides*. Vertical bars represented the standard deviation of growth rates among replicates.

**Figure 2 microorganisms-13-01378-f002:**
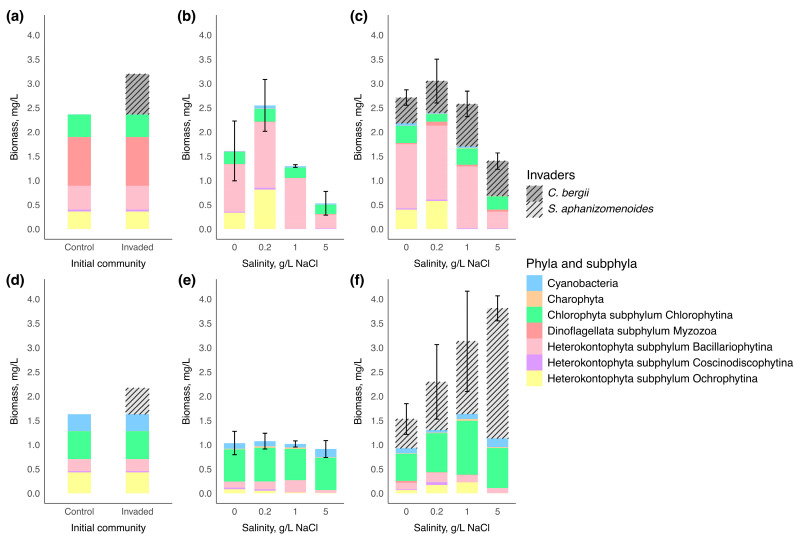
Biomass and composition of phytoplankton communities at the beginning and end of the experiment (n = 3 for each salinity condition). A composition of the phytoplankton community at the beginning of the experiment in (**a**)—water body 1 and (**d**)—water body 2; community response to salinity increases in the control group at the end of the experiment in (**b**)—water body 1 and (**e**)—water body 2; community response to salinity increases in invaded group at the end of the experiment in (**c**)—water body 1 and (**f**)—water body 2. Vertical bars represented the standard deviation of total phytoplankton biomass among replicates.

**Figure 3 microorganisms-13-01378-f003:**
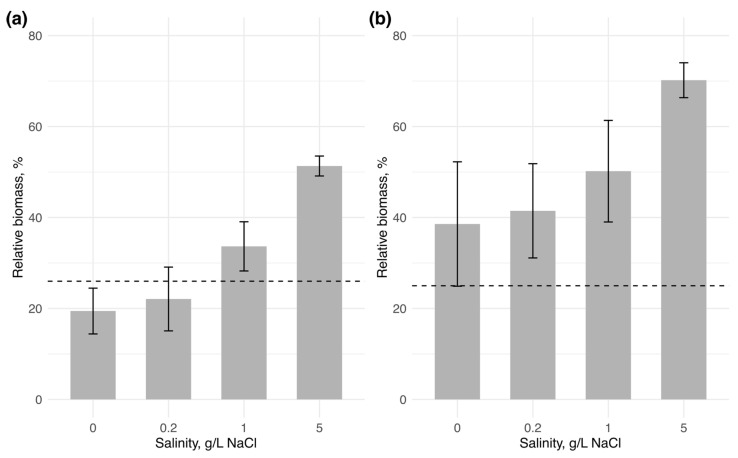
Relative biomass of the invading species to total phytoplankton biomass at the end of the experiment (n = 3 for each salinity condition): (**a**)—*Chrysosporum bergii*, (**b**)—*Sphaerospermopsis aphanizomenoides*. Vertical bars represent the standard deviation across replicates, with the horizontal line indicating the initial relative biomass of the invaders.

**Table 1 microorganisms-13-01378-t001:** List of the studied freshwater cyanobacteria (n = 13) with their origin and year of isolation.

Isolate	Water Body	Country	Year of Isolation
*A. gracile*			
NRC/SIM/2022/D5	Lake Simnas	Lithuania	2022
NRC/JIE/2022/B3	Lake Jieznas	Lithuania	2022
*C. bergii*			
NRC/KYV/2015/E2	Lake Rėkyva	Lithuania	2015
NRC/GIN/2017/B3	Lake Gineitiškės	Lithuania	2017
NRC/GIN/2017/D6	Lake Gineitiškės	Lithuania	2017
NRC/GIN/2017/F6	Lake Gineitiškės	Lithuania	2017
NRC/JIE/2021/B3	Lake Jieznas	Lithuania	2021
*C. issatschenkoi*			
NRC/JIE/2021/D5	Lake Jieznas	Lithuania	2021
NRC/JIE/2021/F5	Lake Jieznas	Lithuania	2021
*S. aphanizomenoides*			
NRC/SIM/2022/A1	Lake Simnas	Lithuania	2022
NRC/HosP/2022/D8	Reservoir Hostivař	The Czech Republic	2022
NRC/HosP/2022/F6	Reservoir Hostivař	The Czech Republic	2022
NRC/HosP/2022/F8	Reservoir Hostivař	The Czech Republic	2022

**Table 2 microorganisms-13-01378-t002:** Results of the Mann–Whitney U test (W), one-way ANOVA (F) and Kruskal–Wallis (χ^2^) evaluating the differences in growth rates under various salinities and among isolates of cyanobacteria. Bold values indicate statistical significance at the *p* < 0.05 level.

Species	Factor	df	Test Statistic	*p*-Value
*A. gracile*	Strain	1	W = 89	0.347
	Salinity	3	F = 59.3	**<0.001**
*C. bergii*	Strain	4	χ^2^ = 15.18	**0.004**
	Salinity	4	F = 40.13	**<0.001**
*C. issatschenkoi*	Strain	1	W = 87	0.410
	Salinity	3	F = 106.5	**<0.001**
*S. aphanizomenoides*	Strain	3	F = 0.39	0.759
	Salinity	4	F = 33.97	**<0.001**

**Table 3 microorganisms-13-01378-t003:** Results of one-way ANOVA (F) and Kruskal–Wallis (χ^2^) tests evaluating biomass differences among taxonomic groups under varying salinity levels in both control and invaded communities. Bold values indicate statistical significance at the *p* < 0.05 level.

Taxonomic Group	Test Statistic	*p*-Value
Water body 1		
Bacillariophytina	F = 16.74	**<0.001**
Charophytina	χ^2^ = 2.02	0.569
Chlorophyta	F = 0.95	0.439
Coscinodiscophytina	F = 5.24	**0.008**
Cyanobacteria	F = 1.53	0.239
Myzozoa	χ^2^ = 2.37	0.499
Ochrophytina	χ^2^ = 19.45	**<0.001**
Water body 1		
Bacillariophytina	F = 5.27	**0.009**
Charophyta	χ^2^ = 5.42	0.144
Chlorophytina	χ^2^ = 1.92	0.590
Coscinodiscophytina	χ^2^ = 16.64	**<0.001**
Cyanobacteria	F = 9.17	**<0.001**
Myzozoa	χ^2^ = 2.05	0.563
Ochrophytina	χ^2^ = 11.15	**0.011**

## Data Availability

The original contributions presented in this study are included in the article/Appendix A. Further inquiries can be directed to the corresponding authors.

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
