# Peer review of "Halotolerance of Phytoplankton and Invasion Success of Nostocalean Cyanobacteria Under Freshwater Salinization"

_microorganisms, 2025, doi:10.3390/microorganisms13061378_

Round 1
Reviewer 1 Report
Comments and Suggestions for Authors
The authors investigated the changes in phytoplankton communities in response to the introduction of alien species and increasing salinity, simulating anthropogenic pollution of aquatic ecosystems. Through well-designed experiments, they convincingly demonstrated that salinity can be a critical factor in invasions and blooms of halotolerant cyanobacteria. The research was conducted at a high methodological standard, with clear visualization of experimental results and comprehensive and easily understandable descriptions of all steps. The discussion of results is also thorough and incorporates relevant and up-to-date literature. This novel and valuable study is clearly relevant to the management of water bodies under conditions of global climate change and increasing harmful cyanobacterial blooms. Minor suggestions are provided below to further enhance the quality of this excellent manuscript:
Line 210-211, 225. Here it is necessary to check the correctness of the taxonomy. Thus, there was an error in the spelling of the genus Oocystis and not all the listed genera belong to the Chlorellales order. It would be more accurate to write that they belong to the class Chlorophyceae.
Author Response
Response to Reviewer 1 Comments
The authors investigated the changes in phytoplankton communities in response to the introduction of alien species and increasing salinity, simulating anthropogenic pollution of aquatic ecosystems. Through well-designed experiments, they convincingly demonstrated that salinity can be a critical factor in invasions and blooms of halotolerant cyanobacteria. The research was conducted at a high methodological standard, with clear visualization of experimental results and comprehensive and easily understandable descriptions of all steps. The discussion of results is also thorough and incorporates relevant and up-to-date literature. This novel and valuable study is clearly relevant to the management of water bodies under conditions of global climate change and increasing harmful cyanobacterial blooms. Minor suggestions are provided below to further enhance the quality of this excellent manuscript.
Answer:
Thank you very much for taking the time to review this manuscript and for your positive evaluation. Please find the detailed response below, with the corresponding revisions highlighted in the re-submitted files.
Comments 1: Line 210-211, 225. Here it is necessary to check the correctness of the taxonomy. Thus, there was an error in the spelling of the genus Oocystis and not all the listed genera belong to the Chlorellales order. It would be more accurate to write that they belong to the class Chlorophyceae.
Response 1: Thank you for pointing this out. We have corrected the spelling of Oocystis in line 219. Regarding the taxonomic classification, our intention was to describe the dominant taxa at the genus level where possible, and at the order level when genus-level identification was not possible. We agree the wording was unclear and have revised the text to: “unidentified species from the Chlorellales order” in lines 220 and 234.
Reviewer 2 Report
Comments and Suggestions for Authors
The manuscript entitled “Halotolerance of Phytoplankton and Invasion Success of Nostocalean Cyanobacteria Under Freshwater Salinization” is devoted to changing phytoplankton community, and biological invasions by newly arrived halotolerant species increasing during freshwater salinization, caused by anthropogenic factors. Authors studied halotolerance of four Nostocalean cyanobacterial species — the native to Europe, Aphanizomenon gracile, and alien Chrysosporum bergii, Cuspidothrix issatschenkoi, and Sphaerospermopsis aphanizomenoides using monoculture experiments under varying NaCl concentrations and microcosm experiments. To my mind this manuscript is corresponding to the aims and scopes of the Microorganisms journal. I am ready to recommend it for publication after corrections, due to the comments below.
- In the abstract, Water body should be specified by indicating the name of the reservoir
- It is necessary to write the purpose of the work more clearly at the end of the introduction
- It is necessary to describe in more detail the process and causes of freshwater salinization caused by anthropogenic factors in the selected reservoirs.
- How confident are the authors in the accuracy of the morphological identification of communities? Today, there are more reliable molecular biological methods for this.
- It is necessary to explain the cultivation temperature of 18 oC. To what extent does it correspond to the selection conditions. Was the plankton cultivated at one temperature? In addition to the salinity, the temperature effect can also affect the invasion. It was studied. This should be discussed somehow in the text. It is not very clear at what temperature the microcosm was worked.
- It seems to me more logical to begin the description of the results with the study of microcosms and then link it with monocultural experiments. This would strengthen the logical connection between the two stages of the work.
- It is not very clear how the accumulation of biomass in grams and the distribution between species were calculated. This should be detailed in the methods
- The discussion of the results, in my opinion, should be more closely related to the geochemistry of lakes and their anthropogenic salinization
Since at the beginning of the work the authors associate the development of invasive species with anthropogenic salinization, this idea needs to be developed in more detail in the conclusion.
Author Response
Response to Reviewer 2 Comments
The manuscript entitled “Halotolerance of Phytoplankton and Invasion Success of Nostocalean Cyanobacteria Under Freshwater Salinization” is devoted to changing phytoplankton community, and biological invasions by newly arrived halotolerant species increasing during freshwater salinization, caused by anthropogenic factors. Authors studied halotolerance of four Nostocalean cyanobacterial species — the native to Europe, Aphanizomenon gracile, and alien Chrysosporum bergii, Cuspidothrix issatschenkoi, and Sphaerospermopsis aphanizomenoides using monoculture experiments under varying NaCl concentrations and microcosm experiments. To my mind this manuscript is corresponding to the aims and scopes of the Microorganisms journal. I am ready to recommend it for publication after corrections, due to the comments below.
Answer:
Thank you very much for taking the time to review this manuscript and for your valuable insights, which have greatly contributed to improving the quality of the work. Please find the detailed response below, with the corresponding revisions highlighted in the re-submitted files.
Comments 1: In the abstract, Water body should be specified by indicating the name of the reservoir.
Response 1: Thank you for pointing this out. However, the water bodies used in this study are artificial reservoirs and do not have official names. For clarity and consistency, we refer to them in the manuscript as “water body 1” and “water body 2”.
Comments 2: It is necessary to write the purpose of the work more clearly at the end of the introduction.
Response 2: Thank you for your comment. We have added a clarifying sentence at the beginning of the final paragraph of the introduction to clearly state the purpose of our study (lines 68–71). We deleted the word “Nostocalean” to avoid repetition (line 71).
Comments 3: It is necessary to describe in more detail the process and causes of freshwater salinization caused by anthropogenic factors in the selected reservoirs.
Response 3: Thank you for your comment. To our knowledge, no salinization event has occurred in the selected reservoirs. These ecosystems were deliberately chosen to assess the effects of experimentally induced salinization on freshwater plankton communities. The conductivity measurements in both reservoirs were close to 0 mS/cm, confirming low natural salinity. In our study, salinization was introduced solely under controlled laboratory conditions by adding NaCl. We specifically selected anthropogenic (man-made) reservoirs, as previous research suggests that such artificial ecosystems may be susceptible to biological invasions.
Comments 4: How confident are the authors in the accuracy of the morphological identification of communities? Today, there are more reliable molecular biological methods for this.
Response 4: Thank you for your comment. We acknowledge the advantages of molecular methods in phytoplankton identification. However, we are confident in the accuracy of our morphological identification, as the analysis was conducted by a help of experienced taxonomists in our laboratory. While not all taxa could be identified to the genus level, this limitation is stated in the manuscript. Additionally, the phytoplankton composition was presented at the phylum and subphylum levels, where taxonomic identification is more reliable and allowed us to minimize the risk of misclassification. We agree that combining microscopy with molecular techniques can enhance taxonomic resolution, but in this study, we considered morphological identification sufficient for our work.
Comments 5: It is necessary to explain the cultivation temperature of 18 oC. To what extent does it correspond to the selection conditions. Was the plankton cultivated at one temperature? In addition to the salinity, the temperature effect can also affect the invasion. It was studied. This should be discussed somehow in the text. It is not very clear at what temperature the microcosm was worked.
Response 5: Thank you for your comment. Both the monoculture and microcosm experiments were performed at a constant temperature of 18 °C, as they were performed in a controlled environment room set to this temperature at the WasserCluster Lunz research facility in Austria. Although the cyanobacterial isolates were taken from the Culture Collection of the Nature Research Centre in Lithuania, temperature of 20 °C were mentioned in the methods section to describe the temperature at which it was isolated and stored there. We agree that temperature can influence invasion dynamics, and while our study focused on salinity as the main environmental stressor, the isolates were acclimated to 18 °C prior to the experiments. We now clarify the cultivation temperature and its consistency across experiments in the revised methods section ”Cyanobacterial isolation and maitenance” (lines 90-96). We removed sentence “The experiments were carried out in the AquaScale lab at WasserCluster Lunz, Austria” in lines 124-125, as it was already mentioned in the section section “Cyanobacterial isolation and maintenance”.
Comments 6: It seems to me more logical to begin the description of the results with the study of microcosms and then link it with monocultural experiments. This would strengthen the logical connection between the two stages of the work.
Response 6: Thank you for your suggestion. However, we chose to present the results starting with the monoculture experiment, as it reflects the actual sequence of our study. The monoculture experiment was performed first to assess the halotolerance of different cyanobacteria, and based on those findings, we selected halotolerant species for the subsequent microcosm invasion experiments under salinity stress. We believe this order best supports the logical flow and rationale of our experimental design.
Comments 7: It is not very clear how the accumulation of biomass in grams and the distribution between species were calculated. This should be detailed in the methods.
Response 7: Thank you for your insight. We have added a reference in the Methods section for the biomass calculation formulas (lines 148-149). We decided to include just the reference instead of the full formulas, similar to how we presented biovolume calculations. However, if you suggest it necessary, we could include the formulas in the revised manuscript.
Comments 8: The discussion of the results, in my opinion, should be more closely related to the geochemistry of lakes and their anthropogenic salinization.
Response 8: Thank you for your suggestion. We agree with that and have made changes. We improved the opening paragraph in the discussion section where we emphasized the selection of small artificial reservoirs, which are considered especially vulnerable to salinization due to their susceptibility to evaporation and exposure to anthropogenic inputs such as urban runoff and agricultural discharge (lines 300-303). In the last paragraph of discussion section, we disucces in more detail what are the consequences of anthropogenic salinization (lines 483-495).
Comments 9: Since at the beginning of the work the authors associate the development of invasive species with anthropogenic salinization, this idea needs to be developed in more detail in the conclusion.
Response 9: Thank you for your comment. We improved the conclusion with more clear explanation of salinization effect on the success of non-native cyanobacteria establishment (lines 498-511).
Reviewer 3 Report
Comments and Suggestions for Authors
Dear Authors,
The manuscript entitled “ Halotolerance of Phytoplankton and Invasion Success of Nostocalean Cyanobacteria Under Freshwater Salinization” has focussed on four species of Nostocalean cyanobacteria, a native one and 4 alien species to assess their halotolerance under varying levels of salinization. Also, one of the main aims was to evaluate the biodiversity shifts in natural freshwater phytoplankton communities from two mesotrophic, oligohaline artificial reservoirs exposed to salinization stress and to examine their susceptibility to cyanobacterial invasion. Results showed that two alien cyanobacteria were halotolerance even under mesohaline conditions and the response of freshwater phytoplankton communities was primarily influenced by the behaviour of the dominant species. This study highlights very important thing in my opinion, and that is the need to consider salinization as a key factor in the management and conservation of freshwater ecosystems.
I think the topic is very interesting, important and suitable for publishing in Microorganisms journal. I think the manuscript is very clear written, so I do not have any comments.
Sincerely,
The reviewer
Author Response
Response to Reviewer 3 Comments
Dear Authors,
The manuscript entitled “ Halotolerance of Phytoplankton and Invasion Success of Nostocalean Cyanobacteria Under Freshwater Salinization” has focussed on four species of Nostocalean cyanobacteria, a native one and 4 alien species to assess their halotolerance under varying levels of salinization. Also, one of the main aims was to evaluate the biodiversity shifts in natural freshwater phytoplankton communities from two mesotrophic, oligohaline artificial reservoirs exposed to salinization stress and to examine their susceptibility to cyanobacterial invasion. Results showed that two alien cyanobacteria were halotolerance even under mesohaline conditions and the response of freshwater phytoplankton communities was primarily influenced by the behaviour of the dominant species. This study highlights very important thing in my opinion, and that is the need to consider salinization as a key factor in the management and conservation of freshwater ecosystems.
I think the topic is very interesting, important and suitable for publishing in Microorganisms journal. I think the manuscript is very clear written, so I do not have any comments.
Sincerely,
The reviewer
Answer:
Thank you very much for taking the time to review this manuscript and for your positive evaluation.
Reviewer 4 Report
Comments and Suggestions for Authors
Dear Authors,
I have read your article, “Halotolerance of Phytoplankton and Invasion Success of Nostocalean Cyanobacteria Under Freshwater Salinization”, and found it smooth and easy to comprehend. The topic is timely and likely to attract the readership of the Microorganisms journal. While the manuscript is overall well-structured, I offer the following constructive comments to support further improvement of the work, based on my perspective. If the authors consider these suggestions, I believe the article will be ready for publication.
Abstract (Lines 27–29): The abstract would benefit from a stronger take-home message. What is the key conclusion or insight that readers should remember from your findings? What new understanding do microalgae physiologists gain from your results and discussion?
Introduction: The introduction is well organized and effectively outlines the state of the art, the motivation behind the work, its novelty, and the central hypothesis.
Discussion Section:
While the discussion is generally of acceptable quality, several improvements are recommended to enhance the clarity and impact of the section:
Although the discussion is divided into subsections, the narrative feels somewhat fragmented. Transitions between comparisons (across studies or between species) are sometimes abrupt and could be smoothed to improve the logical flow.
Avoid redundancies—for example, repeatedly stating the origin of each isolate—unless this detail directly supports a new point or insight.
Strengthen the discussion of the state of the art and articulate more clearly the novel contributions of the present work. How do the findings align with or challenge existing theories? What new knowledge does this study add?
Consider grouping and contrasting findings by ecological relevance (e.g., "brackish invaders vs. freshwater specialists") rather than isolate-by-isolate comparisons, which can become repetitive.
Additionally, the broader ecological and practical implications of the study should be explored more deeply. Expand on the ecosystem-level impacts of salinization and cyanobacterial invasion. What could increase dominance of cyanobacteria mean for food web dynamics, oxygen levels, or water quality?
Discuss potential management implications. How might your findings inform lake management practices or pollution control strategies?
Consider addressing the potential influence of climate change or urbanization on freshwater salinization trends.
Finally, strengthen the conclusion with deeper interpretation. What do these findings reveal about the resilience or vulnerability of freshwater ecosystems under increasing salinity?
Author Response
Response to Reviewer 4 Comments
Dear Authors,
I have read your article, “Halotolerance of Phytoplankton and Invasion Success of Nostocalean Cyanobacteria Under Freshwater Salinization”, and found it smooth and easy to comprehend. The topic is timely and likely to attract the readership of the Microorganisms journal. While the manuscript is overall well-structured, I offer the following constructive comments to support further improvement of the work, based on my perspective. If the authors consider these suggestions, I believe the article will be ready for publication.
Answer:
Thank you very much for taking the time to review this manuscript and for your valuable insights, which have greatly contributed to improving the quality of the work. Please find the detailed response below, with the corresponding revisions highlighted in the re-submitted files.
Comments 1: Abstract (Lines 27–29): The abstract would benefit from a stronger take-home message. What is the key conclusion or insight that readers should remember from your findings? What new understanding do microalgae physiologists gain from your results and discussion?
Response 1: Thank you for your comment. Due to the word limit in the abstract, we are unfortunately unable to expand the explanation. However, we have revised the final sentence of the abstract to better summarize the significance of our study: “Our findings suggest that anthropogenic stressors such as freshwater salinization can alter phytoplankton community and confer a competitive advantage to certain taxa, including widespread alien cyanobacteria, potentially promoting invasion and blooms formation” (lines 27-30).
Comments 2: Introduction: The introduction is well organized and effectively outlines the state of the art, the motivation behind the work, its novelty, and the central hypothesis.
Response 2: Thank you, we appreciate your feedback.
Comments 3: Discussion Section:
While the discussion is generally of acceptable quality, several improvements are recommended to enhance the clarity and impact of the section:
Although the discussion is divided into subsections, the narrative feels somewhat fragmented. Transitions between comparisons (across studies or between species) are sometimes abrupt and could be smoothed to improve the logical flow.
Response 3: Thank you for your comment. We have made some improvements to keep the consitency and the logical flow. First, we have added the first parahraph in section ‘Halotolerance of Nostocalean cyanobacteria’ to agrument why this study was important (lines 314-317). We reordered and improved the connections between results from our study and the comparisons with other studies, change the order of sentences or added additional phrases to sound it more fluent in all section (lines 313-374).
Comments 4: Avoid redundancies—for example, repeatedly stating the origin of each isolate—unless this detail directly supports a new point or insight.
Response 4: Thank you for your suggestion. We have included information about species origin for C. bergii to highlight its broad halotolerance likely linked to its origin (326-327), and for S. aphanizomenoides (341-342), to illustrate that despite its freshwater origin, it can thrive in brackish environments, reflecting its strong adaptive capacity. The origin of other species was removed.
Comments 5: Strengthen the discussion of the state of the art and articulate more clearly the novel contributions of the present work. How do the findings align with or challenge existing theories? What new knowledge does this study add?
Response 5: Thank you for your advice. We improved the sentence about the novelty of the work in the opening paragraph to emphasis the novelty of experiments performed for European strains and invasion experiment under salinity stress lines 310-311. We emphasized the novelty of resilience of freshwater phytoplankton community under high salinities, as the previous studies have shown that already small salinization has a detrimental effect (lines 405-419). And included more infomation at the end of the discussion to present existing theories and how it relates with our work, for example agrees with the common invasion-disturbance theory (lines 469-495).
Comments 6: Consider grouping and contrasting findings by ecological relevance (e.g., "brackish invaders vs. freshwater specialists") rather than isolate-by-isolate comparisons, which can become repetitive.
Response 6: Thank you for your suggestions. We have regrouped the findings starting with halotolerant species C. bergii and S. aphanizomenoides and finish with less tolerant species C. issatschenkoi and A. gracile in lines 322-364.
Comments 7: Additionally, the broader ecological and practical implications of the study should be explored more deeply. Expand on the ecosystem-level impacts of salinization and cyanobacterial invasion. What could increase dominance of cyanobacteria mean for food web dynamics, oxygen levels, or water quality?
Response 7: Thank you for your insight. We have improved the last section of the discussion to overview the broader ecological consequences of salinization, invasion and blooms formation, compared with other important factors as eutrophication and warming (lines 483-489).
Comments 8: Discuss potential management implications. How might your findings inform lake management practices or pollution control strategies?
Response 8: We added short information about importance of monitoring in lines 490-492.
Comments 9: Consider addressing the potential influence of climate change or urbanization on freshwater salinization trends.
Response 9: We mentioned in the final paragraph of discussion the influence of climate change and urbanisation in lines 492-495.
Comments 10: Finally, strengthen the conclusion with deeper interpretation. What do these findings reveal about the resilience or vulnerability of freshwater ecosystems under increasing salinity?
Response 10: Thank you. We revised the conclusion to better highlight the potential resilience and halotolerance of certain taxonomic groups and successful invasions driven by anthropogenic salinization (lines 498-511).
Round 2
Reviewer 2 Report
Comments and Suggestions for Authors
To my opinion, the authors have significantly revised the manuscript and taken into account all my comments. I am ready to recommend the manuscript for publication in this form.
Reviewer 4 Report
Comments and Suggestions for Authors
Dear Authors, the manuscript has been significantly improved, and I now recommend it for publication